# AIBP Regulates Metabolism of Ketone and Lipids but Not Mitochondrial Respiration

**DOI:** 10.3390/cells11223643

**Published:** 2022-11-17

**Authors:** Jun-dae Kim, Teng Zhou, Aijun Zhang, Shumin Li, Anisha A. Gupte, Dale J. Hamilton, Longhou Fang

**Affiliations:** 1Center for Cardiovascular Regeneration, Department of Cardiovascular Sciences, Houston Methodist Research Institute, 6550 Fannin St., Houston, TX 77030, USA; 2Center for Bioenergetics, Houston Methodist Research Institute, 6550 Fannin St., Houston, TX 77030, USA; 3Department of Medicine, Houston Methodist, Weill Cornell Medicine Affiliate, 6550 Fannin St., Houston, TX 77030, USA; 4Weill Cornell Medical College, Cornell University, 407 E 61st St., New York, NY 10065, USA

**Keywords:** AIBP/NAXE, NAD(P)HX epimerase, cardiac tissue, untargeted metabolite profiling, mitochondrial respiration

## Abstract

Accumulating evidence indicates that the APOA1 binding protein (AIBP)—a secreted protein—plays a profound role in lipid metabolism. Interestingly, AIBP also functions as an NAD(P)H-hydrate epimerase to catalyze the interconversion of NAD(P)H hydrate [NAD(P)HX] epimers and is renamed as NAXE. Thus, we call it NAXE hereafter. We investigated its role in NAD(P)H-involved metabolism in murine cardiomyocytes, focusing on the metabolism of hexose, lipids, and amino acids as well as mitochondrial redox function. Unbiased metabolite profiling of cardiac tissue shows that NAXE knockout markedly upregulates the ketone body 3-hydroxybutyric acid (3-HB) and increases or trends increasing lipid-associated metabolites cholesterol, α-linolenic acid and deoxycholic acid. Paralleling greater ketone levels, ChemRICH analysis of the NAXE-regulated metabolites shows reduced abundance of hexose despite similar glucose levels in control and NAXE-deficient blood. NAXE knockout reduces cardiac lactic acid but has no effect on the content of other NAD(P)H-regulated metabolites, including those associated with glucose metabolism, the pentose phosphate pathway, or Krebs cycle flux. Although NAXE is present in mitochondria, it has no apparent effect on mitochondrial oxidative phosphorylation. Instead, we detected more metabolites that can potentially improve cardiac function (3-HB, adenosine, and α-linolenic acid) in the *Naxe^−/−^* heart; these mice also perform better in aerobic exercise. Our data reveal a new role of NAXE in cardiac ketone and lipid metabolism.

## 1. Introduction

NAXE is a secreted protein and, as indicated by its name, binds APOA1 and the APOA1-containing high-density lipoprotein (HDL) [1,2]. NAXE accelerates cholesterol efflux to endothelial cells (ECs), macrophages, microglia, and cancer cells [3,4,5]. In addition, NAXE can regulate lipid rafts independent of cholesterol efflux but dependent on the Rho family member CDC42 [6]. Its control of lipid raft abundance contributes to NAXE regulation of vascular biology [7,8], inflammatory responses [3,9,10,11,12], and anti-tumor function [5]. Systemic NAXE knockout mice show increased retinal angiogenesis in development and greater adult angiogenesis following hindlimb ischemia [8]. In contrast to humans [13], no apparent neurological defects were observed in the global NAXE-deficient mice.

In addition to its role in lipid metabolism, NAXE has epimerase activity required for the repair of a hydrated form of NAD(P)H [14]. The cellular NAD(P)H can be hydrated by glyceraldehyde 3-phosphate dehydrogenase (GAPDH) [15,16,17] or by heat shock (>40 °C) and becomes a hydrated form known as NAD(P)HX [18,19]. The hydrated form inhibits the activity of a variety of dehydrogenases because they can no longer provide electrons [18,19]. Thus, it was hypothesized that there should be a corresponding mechanism to repair the nonfunctional NAD(P)HX [14]. There are R and S forms of NAD(P)HX, but only the latter can be reconverted to a functional NAD(P)H [20,21]. NAXE catalyzes the interconversion of the non-repairable R-form to a repairable S-form NAD(P)HX [14], which produces normal NAD(P)H following a reaction with the dehydratase. Notably, both R- and S-forms of NAD(P)HX can spontaneously turn into cyclic NAD(P)HX, which is no longer amenable to enzymatic repair. Based on its epimerase activity, NAXE has recently been renamed NAD(P)HX epimerase (NAXE). While convincing data from a range of organisms—including bacteria, yeast, plant, and humans—demonstrate that NAXE functions as an epimerase, the in vivo physiological function associated with this enzymatic activity is still unclear. Surprisingly, a recent paper described a moonlighting activity of NAXE in vitamin B6 metabolism that is disengaged from the epimerase function [22].

Since NAXE regulates NAD(P)H metabolism, which is involved in multiple metabolic processes, including glucose metabolism, the pentose phosphate pathway, and the Krebs cycle (also known as the tricarboxylic (TCA) cycle), we conducted an untargeted profiling of metabolites in control and NAXE knockout murine cardiac tissue. We postulated that this unbiased approach would reveal most, if not all, important NAXE-associated metabolites that mirror NAXE functions. Our data showed that the levels of cholesterol and its derivative deoxycholic acid levels are elevated while hexose clusters are reduced in NAXE-depleted mouse cardiac tissue. However, this reduction did not have a significant impact on glucose metabolism, the pentose phosphate pathway, or mitochondrial respiration that requires NAD(P)H. Interestingly, we observed more metabolites enhancing cardiac function in the NAXE knockout mice.

## 2. Materials and Methods

### 2.1. Materials

Culture media for ECs were purchased from Lonza (Houston, TX, USA). Antibody against NAXE was obtained from Novus Biologicals (Littleton, CO, USA). Antibodies against GAPDH and Lamin A/C were obtained from Cell Signaling Technology (Danvers, MA, USA). Antibody against VDAC1 was obtained from Abcam (Boston, MA, USA). Peroxidase-labeled second antibodies were obtained from Jackson ImmunoResearch (Buffalo, NY, USA). NAXE short interference RNA (siRNA) and scrambled control siRNA were purchased from Life Technologies (Carlsbad, CA, USA). The Seahorse XF Cell Mito Stress Test Kit was purchased from Agilent (Santa Clara, CA, USA). Common chemical reagents were purchased from Sigma Aldrich (St. Louis, MO, USA). Reagents for sample preparation and primary metabolite analysis by mass spectrometry included acetonitrile, LCMS grade (JT Baker; Cat. No. 9829-02), isopropanol, HPLC grade (JT Baker; Cat. No. 9095-02), crushed ice pH paper 5-10 (EMD Chem. Inc. Gibbstown, NJ, USA), and nitrogen line with pipette tip 18 MΩ pure water (Millipore, Burlington, MA, USA).

### 2.2. Heart Sample Extraction for LC–MS Analysis

Mice were sacrificed without fasting. We collected 5-month-old murine cardiac tissue, alternating between WT and NAXE knockout to exclude a possible time confound. The samples were prepared at the West Coast Metabolomics Center (University of California, Davis). Twenty mg of cardiac tissue was prepared for metabolite extraction by placement in 1 mL extraction solvent (acetonitrile, isopropanol, and water mixture, 3:3:2) and disrupted by Geno/Grinder homogenizer. Following centrifugation at 2500 rpm for 5 min, 500 µL of supernatant was transferred to a new Eppendorf tube and lyophilized in a Centrivap Cold Trap concentrator (Labconco, Kansas City, MO, USA). The lyophilized samples were then subjected to derivatization.

### 2.3. LC–MS-Based Untargeted Metabolite Profiling and Bioinformatics Analysis

The primary metabolite analyses, including sample preparation, data acquisition, and processing, were performed at the West Coast Metabolomics Center (University of California, Davis). Sample injection and data were acquired as described in Fiehn O. et al. [23]. Bioinformatics analysis of pathways, ChemRICH, and Variable Importance in Projection (VIP) analysis of metabolites were conducted as previously described [24].

### 2.4. NAD(P)HX and NAD(P)H Measurements

Measurements were conducted at the West Coast Metabolomics Center. The murine cardiac tissues were disrupted in 400 µL of 50 mM ammonium acetate (pH 7.0) using steel balls. After centrifugation at 14,000× *g* relative centrifugal force (RCF) for 5 min, the supernatant was transferred to a new tube containing 900 µL of cold chloroform. Following vortex and centrifugation at 14,000× *g* RCF for 2 min, the top aqueous layer was transferred to a new Eppendorf tube and frozen at −80 °C. The samples were lyophilized overnight in the Labconco freeze dryer at a non-speed vacuum mode. The prepared samples were dissolved in 100 µL of 50 mM ammonium acetate (pH 7.0) for LC–MS analysis.

### 2.5. Cell Culture

Human umbilical vein endothelial cells (HUVECs) were obtained from Lonza and cultured under conditions recommended by the manufacturer.

### 2.6. Immunostaining of HUVECs

HUVECs, plated on 0.2% gelatin-coated coverslips, were incubated with 25 μM MitoTracker Deep Red for 1 h, fixed in 4% paraformaldehyde for 15 min at room temperature, permeabilized with Tris-buffered saline (TBS) containing 0.3% Triton X-100 for 10 min, and blocked with TBS–bovine serum albumin for 1 h. After 1 h incubation with primary NAXE antibodies and corresponding secondary antibodies, with 3 × 10 min washes with phosphate-buffered saline after each incubation, the coverslips were mounted using anti-fade mounting medium containing DAPI (VectorLabs, Newark, CA, USA). The slides were air-dried overnight before imaging.

### 2.7. Western Blot

Protein concentration in cell lysates was quantified with Pierce Protein Assay Kit (Thermo Scientific, Rockford, IL, USA). Protein samples (20–40 µg) were resolved by 4–15% MINI PROTEAN precast gels (Bio-Rad, Hercules, CA, USA), transferred to polyvinylidene fluoride membranes, blocked with a 5% skim milk solution, and probed with primary antibodies (1–5 µg/mL). Membranes were incubated with peroxidase-conjugated secondary antibodies (1:10,000 dilution) and visualized using an enhanced chemiluminescent substrate (GE Healthcare, Houston, TX, USA) and Proteinsimple Fluorchem M Imager (San Jose, CA, USA) or Kodak XOMAT films (Rochester, NY, USA).

### 2.8. siRNA Transfection in ECs

ECs were transfected with NAXE siRNA or scrambled control siRNA using Lipofectamine RNAiMAX reagent (Life Technologies, Carlsbad, CA, USA). Cells were cultured to be 60–80% confluent (area) at the time of transfection. Appropriate amounts of transfection reagent and siRNA (20 nM) were diluted in Opti-MEM medium (Life Technologies). Diluted siRNA was added to diluted transfection reagent at a 1:1 ratio. The siRNA/reagent complexes were incubated for 5 min at room temperature and added directly to the cells in culture medium. The culture media were changed to fresh medium the next day and cells were kept in culture for 48 h.

### 2.9. Oroboros Assay

Respiratory function of the mitochondria in postnatal day 5 (P5) neonatal murine cardiac fibers was determined by assessing oxygen flux with a high-resolution Oroboros respirometer (Oroboros Instruments, Innsbruck, Austria) as previously described (16). Briefly, about 5 to 10 mg of cardiac muscle was dissected from the left ventricle, teased and weighed, and loaded into respiratory chambers. The respiratory rates were measured as oxygen flux per mass (nmol/(min*mg)), and all readings were normalized to the dissected muscle bundle mass.

### 2.10. Seahorse Mitochondrial Respiration Assay

Mitochondrial oxidative phosphorylation and glycolysis were analyzed using a Seahorse XF24 Extracellular Flux Analyzer (Agilent) by measuring the oxygen consumption rate (OCR) and extracellular acidification rate (ECAR) in real time. ECs were seeded in 24-well plates designed for XF24 at 20,000 cells per well in complete growth media. The next day, cells were transfected with NAXE or scrambled control siRNA, switched to unbuffered media, and the assay was carried out according to the manufacturer’s protocol. Cells in each well were counted for normalization of OCR using DAPI.

### 2.11. WST-8 Colorimetric Assay

Cellular dehydrogenase activity was assessed by WST-8 colorimetric assay with a CCK-8 Assay Kit (Dojindo Laboratories, Kumamoto, Japan). HUVECs were seeded in a 96-well microplate, transfected with NAXE siRNA or control siRNA, and assayed as suggested by the manufacturer.

### 2.12. Mouse Treadmill Test

The global NAXE knockout mice were generated as previously described [8]. Mice at the age of ~2 months were used for a treadmill test. Mice were weighed, and overweight mice were removed from the test. Finally, 5 control and 6 NAXE KO mice were selected for the treadmill test, as reported in [25], with minor changes. Briefly, 3 consecutive days of training were performed before assessment. During the 5-day experiment, the two groups of mice were subjected to running individually, once a day, with 1 day of rest between two tests. Running distance was recorded and averaged for each mouse.

### 2.13. Statistical Analysis

All data are expressed as mean ± S.E. unless otherwise indicated. Statistical significance was determined using non-parametric Mann–Whitney test. To generate the graphs in figures, we ran an outlier test in Prism (https://www.graphpad.com/quickcalcs/Grubbs1.cfm, accessed on 3 January 2019) and removed the significant outliers, which did not change the statistical significance. * *p* < 0.05; ** *p* < 0.01; and *** *p* < 0.001.

## 3. Results

### 3.1. Untargeted Metabolite Profiling in Murine Cardiac Tissue Highlights the Role of NAXE in Ketone Metabolism

As NAXE is involved in damaged NAD(P)H repair, we speculate that it would regulate the metabolic processes that require NAD(P)H. To test this hypothesis, we carried out an unselected and unbiased profiling of metabolites and obtained a relative abundance of 481, including 151 structurally defined ones, in control and NAXE knockout cardiac tissues (Appendix A). Different bioinformatics analyses were applied to identify the NAXE-regulated metabolites. We sought to discover crucial shared metabolites for both sexes by combining data from male and female mice. Pathway analysis highlighted changes in metabolites associated with ketone body, cholesterol, hexose cluster, and others out of the 151 structurally defined metabolites (Appendix A). VIP analysis was used to calculate the importance feature of individual metabolite from all cohorts and revealed 162 out of 481 total metabolites that may be important NAXE-associated metabolites (Figure 1A, Appendix A). The top 10 VIP metabolites with known chemical structures are shown in Figure 1A. We found a robust increase in ketone metabolite 3-HB levels (Figure 1B). Ketogenesis is associated with adenosine signaling [26], and we found that adenosine content (*p* = 0.13) but not adenosine-associated metabolites inosine or AMP trended to be increased in NAXE knockout mice (Figure 1C). ChemRICH analysis showed that only hexose clusters were significantly reduced in the NAXE knockout cardiac tissue, which is likely due to the increased ketone body 3-HB content (Figure 1D). Plasma glucose levels were similar between control and NAXE knockout mice (Figure 1F). Notably, cholesterol-related metabolites, such as cholesterol and deoxycholic acid, were increased in NAXE knockout cardiac tissue (Figure 1G). However, NAXE absence did not affect the content of squalene, the precursor for cholesterol biosynthesis, nor plasma total cholesterol (TC) levels (Figure 1G,H). Our unbiased analyses identify a novel association of NAXE with ketone metabolism, which parallels reduced glucose levels in the cardiac tissues.

### 3.2. NAXE Knockout Impairs NAD(P)HX Repair in Cardiac Tissue

NAXE is the only known epimerase to catalyze the interconversion of R- and S-NAD(P)HX (Figure 2A), and loss of NAXE orthologue in humans or other organisms increases R- and S-NAD(P)X and cyclic NAD(P)X due to the absence of the epimerase activity [13,14,22,27,28,29]. We measured all the distinct NAD(P)HX levels and found that, as expected, the relative abundance of R-NADHX and cyclic NADHX trended higher in *Naxe^-/-^* cardiac tissue compared with controls, whereas S-NADHX, S-NADPHX, or cyclic NADPHX did not change significantly (Figure 2B). We also did not detect a statistically significant difference in total NAD(P) or NAD(P)H levels between control and *Naxe^-/-^* cardiac tissue (Figure 3A). Thus, in contrast to human fibroblasts or other organisms, absence of NAXE in murine cardiac tissues appears to selectively perturb NADHX but not NADPHX repair. While NAXE knockout does not significantly change cardiac NAD or NADH levels, the NAD levels trend lower, which is consistent with the NAD content in plasma [30]. We further assessed the effect of NAXE on NAD levels in HUVECs. Treatment with GMX1778 (a potent inhibitor of NAD biosynthesis) or heat shock reduced total NAD levels, as expected (Figure 3B). However, while heat shock treatment augmented NAD(P)HX formation, no difference in NAD levels was found in control or NAXE knockdown HUVECs (Figure 3C,D). Our data suggest that NAXE deficiency has no apparent effect on total NAD levels, which agrees with other studies [13,22].

### 3.3. Minimal Effect of NAXE Knockout on Metabolism That Requires NAD(P)H, Including Glucose and Fatty Acid Metabolism, the Pentose Phosphate Pathway, and Mitochondrial Redox Function

Although we did not find a statistically significant change in NAD(P)H content, subcellular changes but not global NAD(P)H content changes may occur and influence certain metabolic events. We thus probed for changes in intermediate metabolites involved in glucose metabolism, the pentose phosphate pathway, and mitochondrial respiration that require NAD(P)H (Figure 4A, Figure 5A and Figure 6A). However, we did not find any significant changes in glucose-6-phosphate, fructose-6-phosphate, glycerate-3-phosphate, or aspartic acid, which are produced in glucose metabolism (Figure 4B). Only lactic acid levels were reduced in the *Naxe^-/-^* heart (Figure 4B). Glucose-6-phosphate, a metabolite of glucose metabolism, is also used in the oxidative phase of the pentose phosphate pathway, while ribulose-5-phosphate and ribose-5-phosphate are used in the non-oxidative phase. None of them were affected in NAXE knockout cardiac tissue (Figure 5B). All the Krebs cycle metabolites detected—including citrate, aconitate, iso-citrate, α-ketoglutarate, fumarate, succinate, and malate—were comparable in the control and *Naxe^-/-^* animals (Figure 6A,B).

We then assessed free fatty acid content and found that C22 to C16 fatty acid levels were comparable in control and NAXE knockout myocardium (Figure 7). α-linolenic acid levels tend to be elevated by NAXE deficiency (1.39-fold); however, the increases did not reach statistical significance (*p* = 0.42). We further compared the content of acetyl-CoA and acetoacetate-generating amino acids (Figure 8A) and found that lysine levels tended to be increased while phenylalanine levels were reduced in the NAXE knockout cardiac tissue (Figure 8B,C). Thus, NAXE deficiency has no significant effect on NAD(P)H involved in the cardiac metabolism of glucose, pentose phosphate, and amino acids or flux of the Krebs cycle.

### 3.4. A Subset of Endogenous NAXE Is Localized in the Mitochondria of Cardiomyocytes

Overexpressed full-length NAXE localizes to the mitochondria [31], but the cellular distribution of endogenous NAXE remains unknown. To probe the localization of NAXE in vivo, we purified recombinant human NAXE proteins and generated polyclonal antibodies for NAXE. Western blot analysis of whole-cell lysates using this antibody revealed a single band with a molecular mass of 28 kDa (Appendix A), suggesting that the antibody we generated is specific for NAXE. Immunostaining of cardiomyocytes showed a strong signal of endogenous NAXE by the NAXE antibody, and some of the signal colocalized with mitochondria that were marked by MitoTracker Deep Red (Figure 9A). In addition, a substantial amount of NAXE was localized in the mitochondria of HUVECs (Figure 9B). Notably, pronounced colocalization of NAXE with mitochondria occurred in the perinuclear region of HUVECs. Our results indicate that NAXE is present in different subcellular locations, including mitochondria.

### 3.5. NAXE Shows No Significant Impact on Mitochondrial Respiration

We explored the role of NAXE in mitochondrial redox function using murine cardiac myofiber, which is enriched with mitochondria and generates most, if not all, of the ATP via mitochondrial oxidative phosphorylation [32,33,34]. We used neonatal mouse cardiac tissue for the mitochondrial function analysis because neonatal cardiomyocytes prefer to use carbohydrates as an energy source, unlike adult cardiac tissue [35,36]. Since in vitro cell culture conditions may change the intrinsic property of cells and interfere with subsequent analysis of mitochondrial function [37], to minimize the influence of cell culture, we used the Oroboros assay to measure mitochondrial function using freshly prepared cardiac myofibers dissected from the left ventricle, thereby avoiding cell cultures needed for the Seahorse assay. In this protocol, the background respiration is recorded prior to any perturbation. Basal respiration is calculated by measuring the initial oxygen consumption and subtracting the residual oxygen consumption upon addition of the electron transport complex I and III inhibitors, rotenone (Rot), and antimycin A (AA), respectively (Figure 6C). The decreased oxygen consumption obtained upon the addition of oligomycin (OM), which inhibits ATP synthase, determines coupled mitochondrial respiration—the oxygen consumption used to generate ATP (Figure 6C). Maximal respiration is determined by injection of carbonyl cyanide p-trifluoromethoxyphenylhydrazone (FCCP), a versatile protonophore that uncouples ATP generation from oxygen consumption allowing transport of protons across the mitochondrial inner membrane instead of through the proton channel of ATP synthase (Figure 6C). Following the sequential injection of OM, FCCP, and then Rot and AA to inhibit different mitochondrial complex functions, we were unable to detect significant differences in mitochondrial respiration between *Naxe^−/−^* and control cardiac tissue (Figure 6D).

We also analyzed the effect of NAXE on HUVEC mitochondrial respiration using the Seahorse assay. siRNA-mediated gene knockdown was applied to reduce NAXE expression (Appendix A). As in cardiac tissue, NAXE deficiency in HUVECs showed no apparent effect on mitochondrial oxidative phosphorylation as measured by seahorse assay (Appendix A) or mitochondrial NAD(P)H-dependent reductase assay (Appendix A). Thus, in ECs that generate the majority of their ATP using mitochondria-independent glycolysis [38] or cardiac myofibers that rely extensively on mitochondria-generated energy, NAXE does not have an appreciable role in cellular mitochondrial respiration.

### 3.6. NAXE Knockout Mice Show Greater Ambulatory Performance on a Treadmill Test

Given that greater adenosine content appears to have a cardioprotective effect and an increase in 3-HB levels can provide easily accessible fuel to generate acetyl-CoA for mitochondrial respiration, we examined the ambulatory ability of NAXE knockout mouse on a treadmill. We used an experimental protocol that determines their aerobic performance. The treadmill speed was initially set at 10 m/min and gradually increased every 3 min, reaching the final speed of 36.6 m/min at 48 min, as reported [25]. As anticipated, the NAXE null mice ran a longer distance compared with the WT mice (Figure 10; 789.87 vs. 609.67 m). The data suggest that NAXE deficiency increases murine endurance in a treadmill test.

## 4. Discussion

### 4.1. A Novel Role of NAXE in Ketone Metabolism

Using untargeted metabolite profiling, our studies uncover important NAXE-associated metabolites—in particular, ketone metabolism and cholesterol metabolism-associated pathways (Figure 1). Interestingly, although hexose clusters are decreased in NAXE-null cardiac tissue (Figure 1D,E), metabolites associated with glucose, the pentose phosphate pathway, free fatty acids, or the Krebs cycle did not vary (Figure 4B, Figure 7B, Figure 8B and Figure 9B). Consistent with this, Oroboros analysis detected similar responses in control and *Naxe ^-/-^* cardiac muscle fibers. As the plasma glucose levels were comparable between control and NAXE knockout mice (Figure 1F), presumably, higher 3-HB content impairs or compensates hexose uptake. Lactic acid is the only metabolite derivative of glucose metabolism that trends toward a reduction. Since lactate can be used as cellular fuel [39], it is possible that lactate usage is increased in the NAXE knockout mice.

Acetyl-CoA is an essential substrate for mitochondrial oxidative phosphorylation, ketogenesis, and steroid biosynthesis; it is derived from glucose metabolism, fatty acid oxidation, or amino acid metabolism. Due to technical incompatibility, our metabolite profiling did not detect acetyl-CoA and carnitine, which are required for long-chain fatty acid import into the mitochondria for β-oxidation.

Previous studies show that NAXE deficiency contributes to greater R-, S-, and cyclic NADHX and NADPHX levels; however, our data from murine cardiac tissue indicate that NAXE deletion selectively affects R- and cyclic NADHX content but not NADPHX variants. This may be due to the fact that the adult heart uses fatty acids as a major fuel source to generate energy, producing abundant NADH during β-oxidation that is amenable to damage.

NAD levels are also reduced in the NAXE knockout heart, which is consistent with our findings of metabolites in the plasma [30]. The reduced NAD levels are probably the cause of the red, flaky skin of Pellagra disease in patients with NAXE mutation [13,40].

NAXE absence increases cellular lysine, which is a potential source for greater 3-HB levels. In addition, increased adenosine levels are found in the NAXE knockout murine heart. As ATP is the major precursor for adenosine, we speculate that more ketone body raises ATP, thereby elevating adenosine levels. NAXE deficiency in the brain also analogously changes many of these metabolites, as seen in the cardiac tissue (data not shown). Among these metabolites, adenosine [41] and ketone body [42] can influence neuronal activities. Whether and how these metabolites contribute to neurometabolic defects in patients harboring NAXE mutation is unknown.

### 4.2. What Is the Possible Source for Elevated Ketone Body 3-HB in NAXE Knockouts?

Since 3-HB is an immediate fuel for the myocardium, it can be absorbed rapidly for energy production. It is unclear why 3-HB levels are built up instead of being utilized. We speculate that the influx of acetyl-CoA is saturated for Krebs cycles, and the excess acetyl-CoA is converted to 3-HB. Current wisdom posits that 3-HB is derived from acetyl-CoA or acetoacetyl-CoA. Glycolysis, fatty acid oxidation, or isoleucine metabolism generate acetyl-CoA, while leucine, lysine, phenylalanine, tyrosine, and tryptophan produce acetoacetyl-CoA. Since NAXE knockout did not change the intermediate metabolites of glycolysis or the aforementioned amino acids, except that lysine tends to be increased while phenylalanine is decreased, it is likely that fatty acid β-oxidation-mediated generation of acetyl-CoA is increased. Consistent with this hypothesis, NAXE LOF mutation in zebrafish contributes to mild hyperlipidemia and lipid deposition in cardiac tissue [43]. NAXE is a highly conserved protein from zebrafish to humans (68% identity), and NAXE is a positive hit in a screening for genes associated with liver dysfunction and hepatosteatosis. Since, in mice, there is no difference in plasma TC levels between control and NAXE knockouts, it is possible that hyperlipidemia observed in zebrafish is due to greater triglyceride levels. It is tempting to speculate that NAXE knockout increases lipolysis, where fatty acid β-oxidation-derived acetyl CoA will be augmented and converted into 3-HB.

### 4.3. NAXE and Mitochondrial Function

Although we did not identify a role of NAXE in mitochondrial respiration, NAXE does regulate other mitochondrial functions, such as mitophagy [44,45], mitochondrial morphology, and ATP production [46]. The pathologies associated with these NAXE-orchestrated functions include atherosclerosis [44,45] and retinal neuronal dysfunction [46].

However, many of the pathologies resulting from NAXE mutation in patients suggest a mitochondrial complex I dysfunction [13,44]. It is unclear why this deficiency did not manifest an attenuation of mitochondrial redox function. However, if this is indeed the case, rapamycin—the mTOR inhibitor—may be a candidate for therapy because it improves the neural defects associated with complex I deficiency [47].

### 4.4. NAXE Deficiency and Cardiovascular Function

Since NAXE accelerates cholesterol efflux, it is not surprising that loss of NAXE augments cholesterol content in cardiac tissue. De novo cholesterol synthesis is not changed, as cholesterol precursor squalene levels are similar in control and NAXE knockout cardiac tissue. However, behenic acid (22:0), the consumption of which elevates plasma LDL-C but not HDL-C levels [48], is also increased in NAXE null cardiac tissue. It is possible that behenic acid also contributes to greater cholesterol content in cardiac tissue.

Deoxycholic acid, one of the secondary bile acids, is synthesized by gut flora from cholic acid secreted by the liver and usually functions as a sequestrant for cholesterol reabsorption or fecal elimination [49]. Deoxycholic acid secretion is atheroprotective, and its levels inversely correlate with coronary artery disease [50]. Interestingly, emerging evidence suggests that deoxycholic acid also acts as a ligand to regulate cardiovascular function. For instance, deoxycholic acid exerts a vasodilatory function by promoting Ca^2+^-dependent NO release in human and bovine ECs [51]. Furthermore, deoxycholic acid can activate nuclear farnesoid X receptor (FXR) for subsequent transcriptional regulation [52,53]. Importantly, FXR is expressed in the cardiovascular system, including cardiac and arterial tissues [54,55]. Increased expression of deoxycholic acid transporters could be responsible for the higher levels of deoxycholic acid in NAXE-deficient cardiac tissue.

It is worth noting that NAXE knockout mice show greater exercise capacity in the treadmill test, which is likely due to the concerted effects of more adenosine [56] and 3-HB [57]; the two confer cardio-protection or a versatile energy source. In future studies, we will determine the cell- and non-cell autonomous contributions to changes in these metabolites.

### 4.5. Selective Role of NAXE in Cardiac NADHX Repair

NAXE subcellular distribution implies that it has a role in mitochondrial function. Indeed, reduced complex I activity and pyruvate oxidation were reported in a patient muscle sample with NAXE loss-of-function (LOF) [13]. However, our data from murine cardiac tissue demonstrate that total NAD(P)H levels are not significantly changed with NAXE deficiency. Our study is in accordance with those in Arabidopsis or human cells, documenting no effect on NAD(P)H abundance in the absence of NAXE or NAD(P)HX dehydratase [13,29]. It may be that the NAD(P)HX pool is marginal compared with the normal NAD(P)H content. Indeed, total NADHX accounts for only 1% of total NADH content in cells [58]. It is possible that NAXE plays a yet-to-be-identified role in mitochondria that is less dependent on NAD(P)H levels. For instance, a metabolites-based screen analysis suggests that NAXE might play a role in ADP modification by serving as an NAD donor [59].

We also investigated the role of NAXE in cardiac myofibers and EC mitochondrial respiration. We did not detect any apparent differences in mitochondrial respiration between NAXE-deficient and control cells with a panel of chemical treatments that inhibited the distinct mitochondrial complexes. Moreover, even with heat shock treatment prompting the physiological role of NAXE as an epimerase, we failed to detect substantial changes in mitochondrial oxygen consumption in NAXE-deficient ECs (data not shown). Future studies to dissect the physiological role of NAXE in mitochondria are warranted. Patients with NAXE LOF mutations or a mutation that results in diminished NAXE expression manifest neurodegenerative disorders and skin lesions, which were attributed to the accumulation of cyclic NAD(P)HX and the associated toxic effects [13]. However, in NAXE knockout mice, we did not observe apparent morphological defects under normal conditions, which agrees with the no overt growth defect in NAD(P)H epimerase mutants of *E. coli*, yeast, or the plant Arabidopsis [22,27,28,29,60,61]. This discrepancy in the NAXE LOF phenotype may be due to the existence of one or more functional counterparts. However, even if there is such a functional homolog of NAD(P)H epimerase, it is not as effective as NAXE because markedly increased NADH(P)X and cyclic NADH(P)X were found in the NAXE LOF mutant [13]. Another possibility is that all these species may be more tolerant than humans to the toxic effect of cyclic NAD(P)HX.

## 5. Conclusions

In summary, we applied untargeted metabolite profiling, two different cell types, and diverse methods to explore cellular NAXE function. Although multiple lines of evidence show that NAXE functions as an NAD(P)HX epimerase [13,14,22,27,28,29], the in vivo physiological significance associated with this enzymatic activity is yet to be determined. A recent study suggests that NAXE co-evolves with genes governing vitamin B6 metabolism and shows that NAXE orthologue regulates vitamin B6 synthesis in bacteria [58]. However, vitamin B6 is usually not synthesized in mammals [62]. Further studies are warranted to reconcile cholesterol metabolism with the NAXE epimerase function.

## Figures and Tables

**Figure 1 cells-11-03643-f001:**
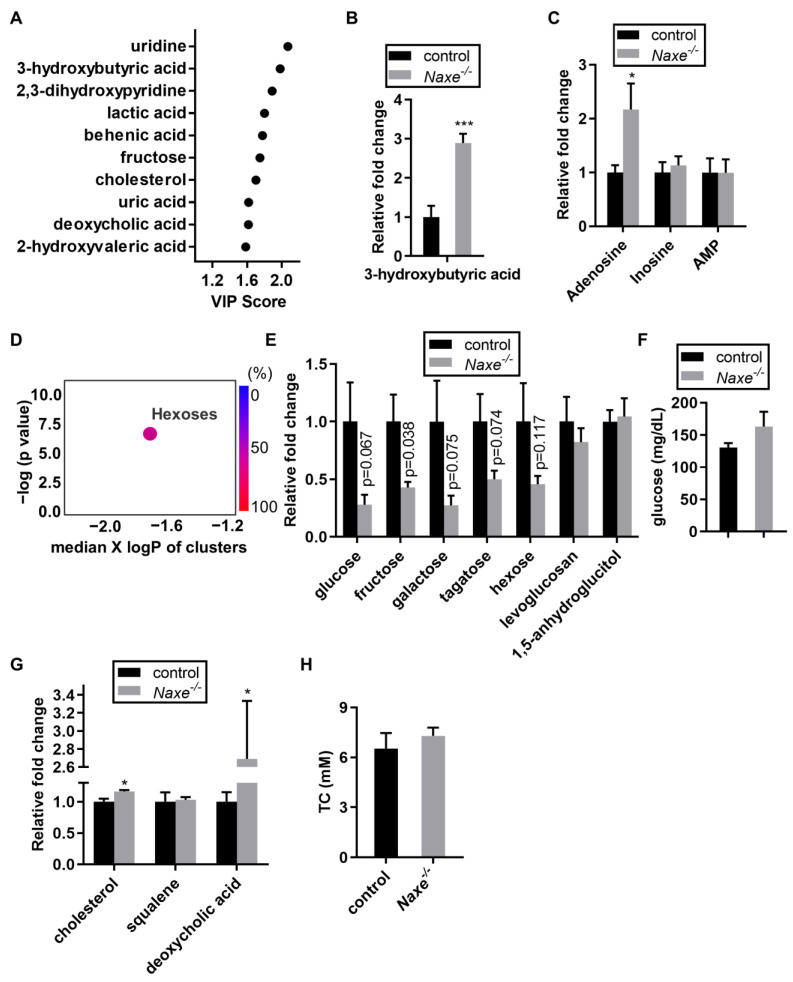
Untargeted profiling identified important metabolites associated with NAXE functions. (**A**) Top 10 VIP metabolites that may be associated with NAXE functions. (**B**) The relative abundance of ketone body 3-hydroxybutyric acid and (**C**) adenosine-associated metabolites in control and NAXE knockouts. (**D**) ChemRICH set-enrichment statistics plot shows enrichment of hexose cluster metabolites associated with NAXE function in the heart. Each node reflects a significantly altered cluster of metabolites. Enrichment *p*-values are given by the Kolmogorov–Smirnov test. Node sizes represent the total number of metabolites in each cluster set. The node color scale shows the scale of metabolite changes in NAXE knockout compared with WT control samples. (**E**), The relative abundance of seven components of the hexose cluster. (**F**), Plasma glucose measurement. (**G**), Relative cholesterol, squalene, and deoxycholic acid abundance in control and NAXE knockout mice. (**H**), Total cholesterol (TC) measurement in the plasma of control or NAXE knockout mice. * *p* < 0.05; *** *p* < 0.001; n = 6 mice per group.

**Figure 2 cells-11-03643-f002:**
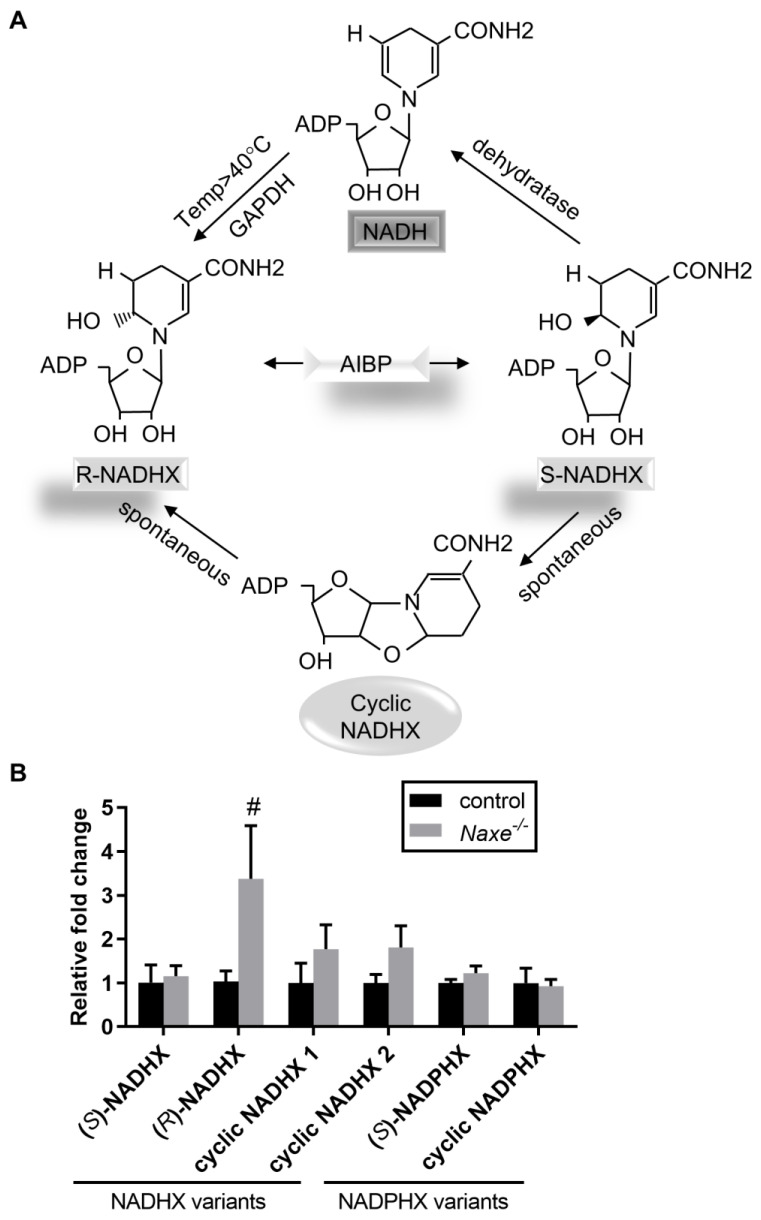
Effect of NAXE on NAD(P)HX content in cardiac tissue. (**A**) NAXE mediates the interconversion of R- and S-NAD(P)X, which can either be repaired to normal NAD(P)H or become cyclic NAD(P)HX. (**B**) LC–MS-based measurements of distinct NAD(P)HX in control or NAXE knockout cardiac tissue. Mean ± SE; # *p* = 0.087; n = 6 mice per group.

**Figure 3 cells-11-03643-f003:**
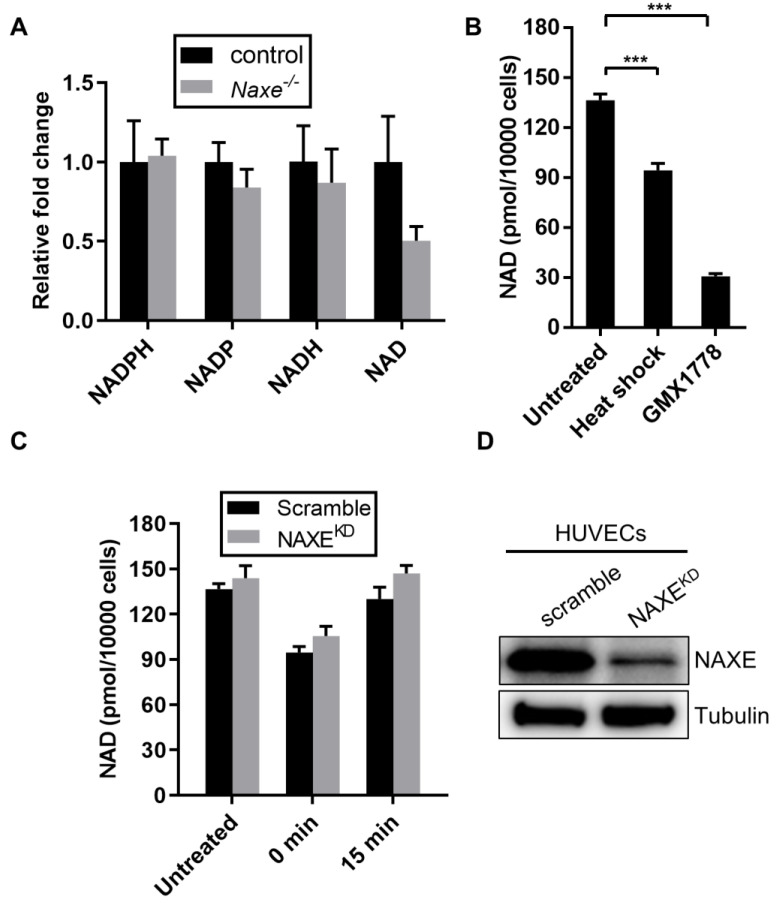
Effect of NAXE deficiency on NAD(P) and NAD(P)H abundance. (**A**) Relative abundance of NAD(P) or NAD(P)H in control or *Naxe^−/−^* cardiac tissue. (**B**) HUVECs plated in a 96-well microplate were subjected to heat shock (1 h at 45 °C) or to overnight GMX1778 treatment (positive control) to induce NADH damage, and total NAD levels were measured. GMX1778 is an inhibitor of nicotinamide phosphoribosyltransferase. (**C**) HUVECs were transfected with NAXE siRNA or scramble control siRNA and subjected to heat shock as in B and allowed to recover for indicated times, and total NAD levels were measured. (**D**) NAXE knockdown was verified using Western blot. Mean ± SE; *** *p* < 0.001; n = 6.

**Figure 4 cells-11-03643-f004:**
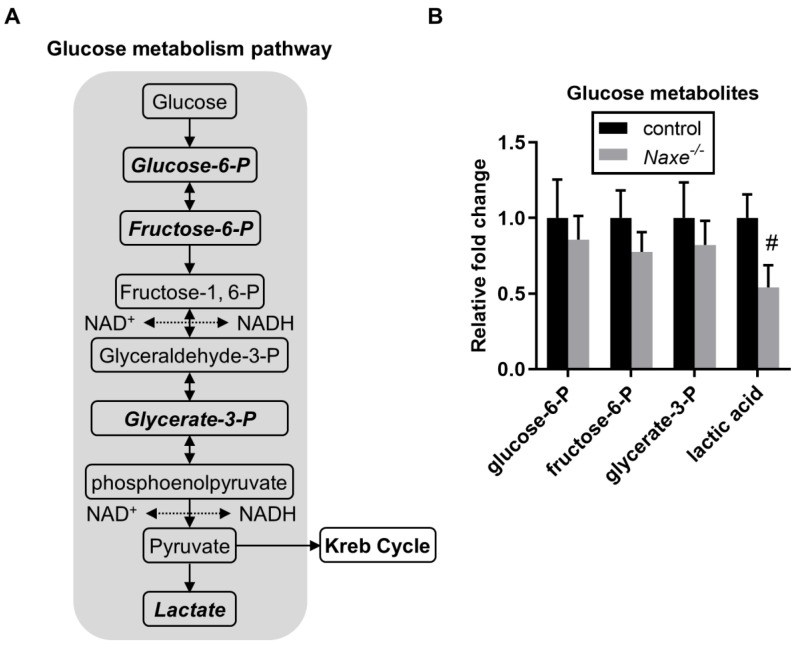
Effect of NAXE deletion on glucose metabolism. (**A**) Scheme depicting the glucose metabolism pathway. (**B**) Relative abundance of detected glucose metabolites. Mean ± SE; # *p* = 0.057; n = 6 mice per group. Detected metabolites are indicated in bold and italic letters.

**Figure 5 cells-11-03643-f005:**
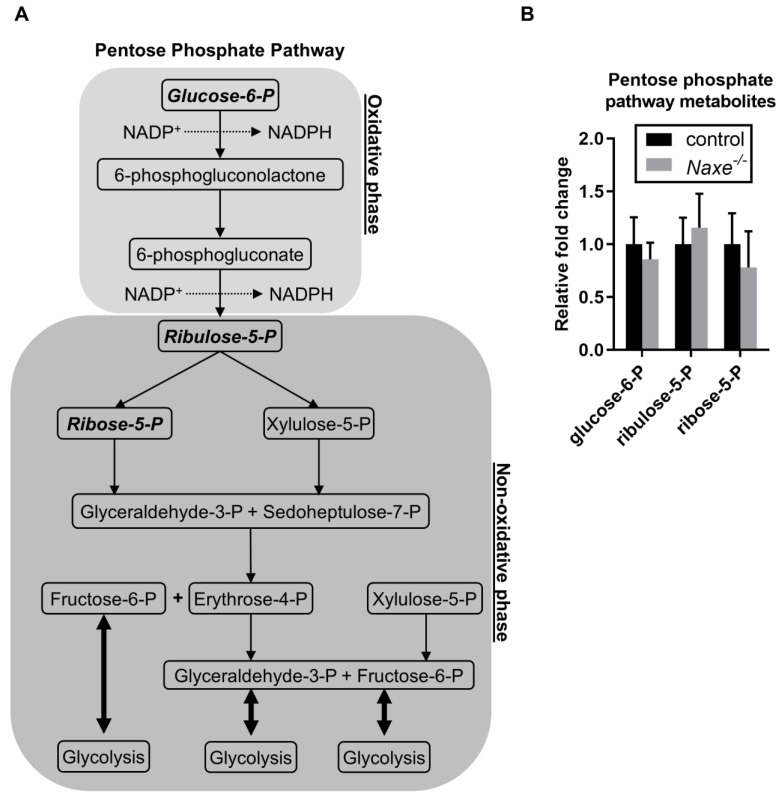
Effect of NAXE knockout on pentose phosphate pathway metabolism. (**A**) Scheme depicting pentose phosphate pathway metabolism. (**B**) Relative abundance of detected pentose phosphate pathway metabolites. n = 6 mice per group. Detected metabolites are indicated in bold and italic letters. Mean ± SE; n = 6.

**Figure 6 cells-11-03643-f006:**
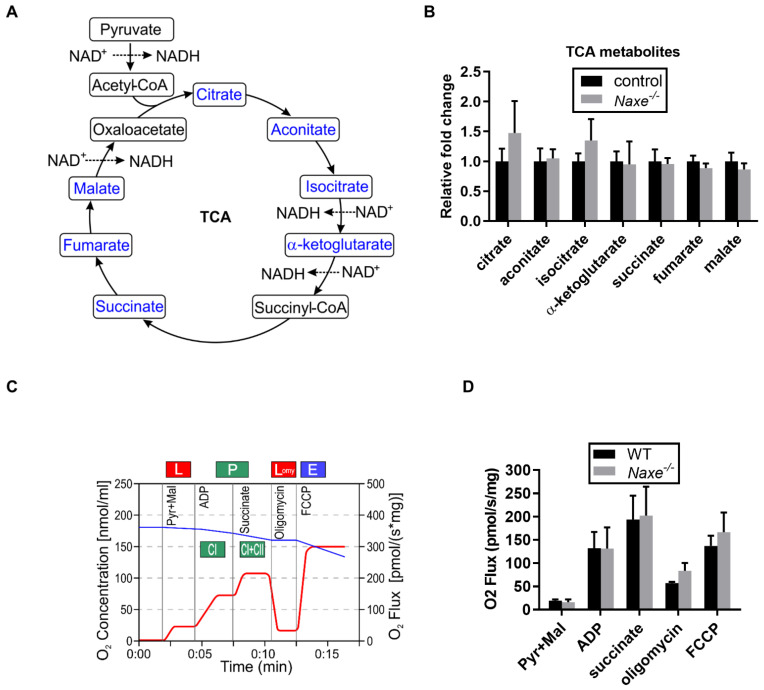
Effect of NAXE on mitochondrial function of murine cardiac tissue. (**A**) Scheme illustrating Krebs cycle in the mitochondria. (**B**) Relative abundance of tricarboxylic acid (TCA) metabolites. (**C**) Schematic diagram showing the protocol and analysis of key parameters of the Oroboros assay. L: leak, P: phosphorylation, Lomy: leak with oligomycin, E: electron transfer, CI: complex I phosphorylation, CI + CII: complex I and II phosphorylation. (**D**) Cardiac myofibers were harvested from P5 neonatal mice, single cell suspension prepared, and Oroboros assay performed to assess the role of NAXE in the mitochondrial function of murine cardiac tissues. The oxygen (O_2_) flux was normalized to the weight of the assayed cardiac tissues. Pyr + Mal: pyruvate–malate. Mean ± SE; n = 6 mice per group for (**B**) and n = 4 mice per group for (**D**). Detected metabolites are indicated in blue.

**Figure 7 cells-11-03643-f007:**
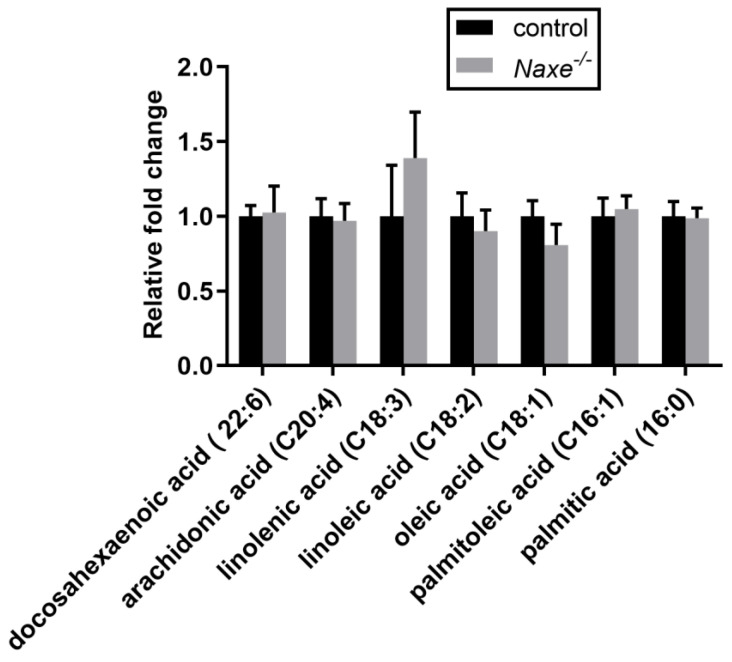
Effect of NAXE knockout on fatty acid metabolism. Relative abundance of detected free fatty acids. Mean ± SE; n = 6 mice per group.

**Figure 8 cells-11-03643-f008:**
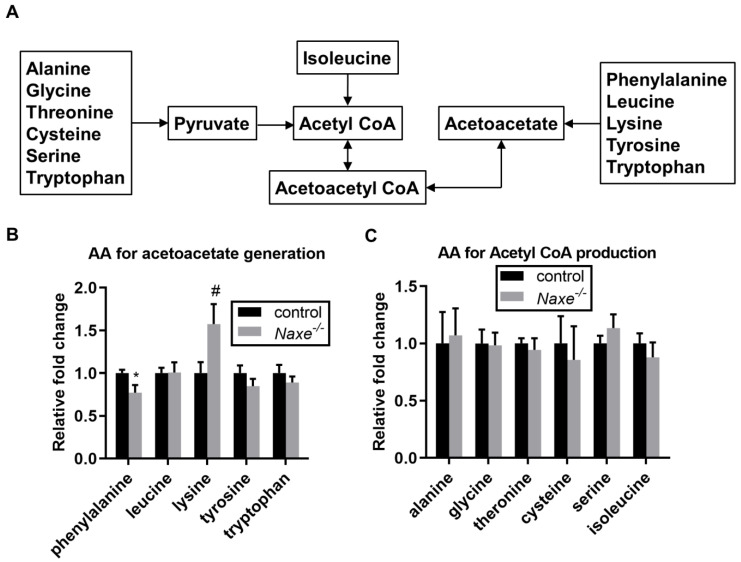
NAXE effect on the content of amino acids that are associated with acetyl-CoA generation. (**A**) Scheme illustrating acetyl-CoA or acetoacetyl-CoA-generating amino acid metabolism. (**B**,**C**) Relative abundance of detected amino acids that can be used for acetyl-CoA or acetoacetyl CoA-production. Mean ± SE; n = 6 mice per group. AA: amino acids. * *p* < 0.05; # *p* = 0.056.

**Figure 9 cells-11-03643-f009:**
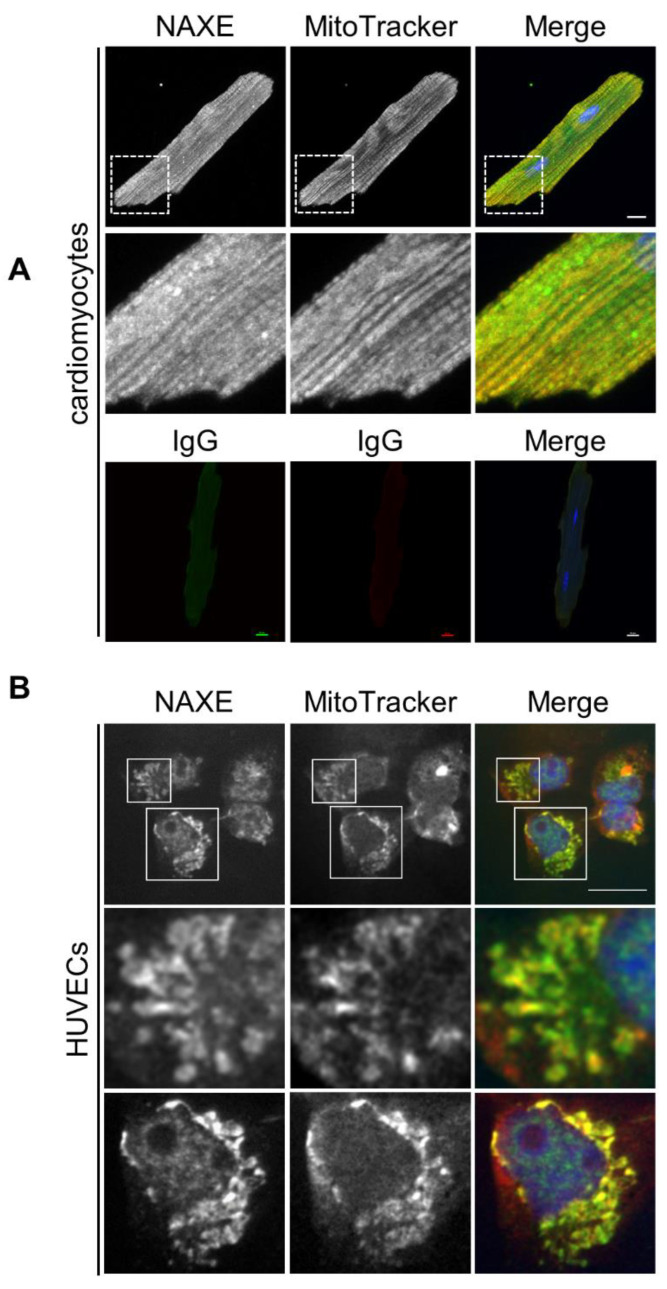
Subcellular localization of NAXE. (**A**) Cardiomyocytes were fixed and stained with our homemade NAXE Abs (green) or with control secondary antibody alone. DAPI staining (blue) was employed to visualize the cell nuclei, and the images were captured using a Leica fluorescence microscope. (**B**) HUVECs were incubated with 25 µM MitoTracker Deep Red for 1 h; then, they were fixed and immunostained with NAXE Abs (green). The enlarged images traced by the white lines are shown below. Note that vesicle-like NAXE staining is found inside the cytosol. Scale Bar: 25 µm.

**Figure 10 cells-11-03643-f010:**
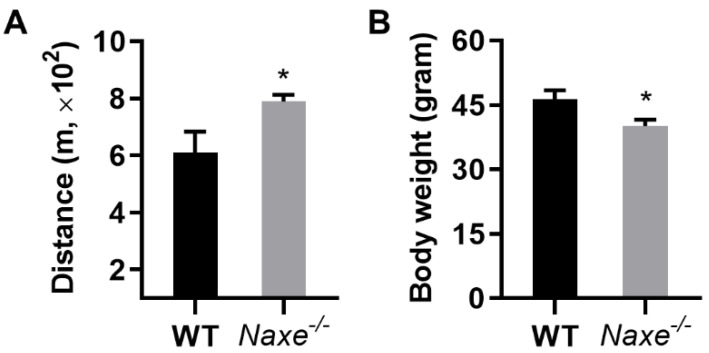
(**A**) Ambulatory performance of control and NAXE knockout mice. The mice were trained for 3 consecutive days before the endurance assessment. Mouse running was performed and examined individually, with one round of test for a total of 5 control and 6 NAXE knockouts each day. The mice were allowed to rest for one day between two tests. Running distance was recorded and averaged for each mouse. (**B**) Body weight of WT and NAXE knockout mice. Mean ± SE; * *p* < 0.05.

## Data Availability

The datasets used and/or analyzed for this study are available upon request from the corresponding author.

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
