# Peer review of "AIBP Regulates Metabolism of Ketone and Lipids but Not Mitochondrial Respiration"

_cells, 2022, doi:10.3390/cells11223643_

Round 1

Reviewer 1 Report

This is an interesting manuscript that evaluates the metabolic and functional consequences of APOA1 binding protein (AIBP) deletion in the cardiovascular system.  The authors present several novel findings that are of interest to investigators in this field.  Several suggestions include:

-        - An expanded description of the knockout mice would be helpful.  Were these global or cardiomyocyte-specific knockout animals?  If these have been published previously, a citation in the Methods section would be beneficial.

-        -The investigators alternate between AIBP and NAXE in the manuscript.  It would be helpful to use one nomenclature.

-        -For some of the experiments, the investigators assessed cardiac tissue.  Was this whole hearts or specific portions of the heart?

-        -The investigators indicate these were combined data from male and female animals.  Were the two genders represented equally in the samples?

-        -In some of the experiments (Figure 1, for instance), the investigators alternate between calling the samples cardiac tissue and cardiomyocytes.  If the samples are not purified cardiomyocytes, stick to cardiac tissue.

Author Response

This is an interesting manuscript that evaluates the metabolic and functional consequences of APOA1 binding protein (AIBP) deletion in the cardiovascular system.  The authors present several novel findings that are of interest to investigators in this field. 

Response: We thank this reviewer for the complimentary comments!

Several suggestions include:

  1. An expanded description of the knockout mice would be helpful.  Were these global or cardiomyocyte-specific knockout animals?  If these have been published previously, a citation in the Methods section would be beneficial.

Response: As this reviewer suggested, we added description of the of global AIBP knockout mice as following: “In systemic NAXE knockout mice, in contrast to humans [13], other than increases retinal angiogenesis in development and facilitates adult angiogenesis and blood flow recovery following hindlimb ischemia [8], no apparent neurological defects were observed.  This sentence is in page 2, lines 45 – 47 of the revised manuscript. So far, no cardiomyocyte-specific knockout animal is available. As suggested, we have put this reference (Mao R et al., Circ Res 2017) in the Method section (page 4, line 160 of the revised manuscript).

  1. The investigators alternate between AIBP and NAXE in the manuscript.  It would be helpful to use one nomenclature.

Response: Thank you for this suggestion. Other than the title and initial introduction, we have changed the name to NAXE in the manuscript. While calling it NAXE conforms to the current nomenclature, the name AIBP seems to be more biologically relevant in this manuscript. Thus, we chose to keep AIBP in the title.

  1. For some of the experiments, the investigators assessed cardiac tissue.  Was this whole hearts or specific portions of the heart?

Response: For all the metabolite measurements, we used the whole cardiac tissue. But to perform the Oroboros analysis (Fig. 9D), we dissected the cardiac muscle fibers from the left ventricle and used them for this analysis. Please also see Response 5.

  1. The investigators indicate these were combined data from male and female animals.  Were the two genders represented equally in the samples?

Response: We used 4 female and 2 male AIBP knockout mice. To generalize our conclusions on AIBP, we carried out the analyses by combining the datasets of both genders.

  1. In some of the experiments (Figure 1, for instance), the investigators alternate between calling the samples cardiac tissue and cardiomyocytes.  If the samples are not purified cardiomyocytes, stick to cardiac tissue.

Response: This is a good point. As elaborated above in Response 3, we used the whole cardiac tissue for the metabolite measurements and used the cardiac muscle fibers for mitochondrial respiration analysis. In the revised manuscript, we changed “cardiomyocytes” to “cardiac muscle fiber” (p17, line 349 of the revised manuscript), cardiac myofiber (p19, line 430)” or “cardiac tissue” (p17, line 359 and p18, lines 403&404) when it is applicable.

Reviewer 2 Report

Well written manuscript and definately well presented.

I would ask for some clarifications regarding data analysis in the 3 following points.

1. An examination of the supplementary material shows that the estimated p values were based on two sided independent samples t tests assuming equal variances. Still, a descriptive approach shows that the variances in the two groups, especially in somes indices, including the ones that statistically significant differences were detected, are rather different.  Based on which criterion did the authors choose to carry out an analysis assuming equal variances in, unexceptionally, all differences examined?

2. Since the sample size would not allow for reliable normal normal distribution testing, were non parametric tests considered? And if so why did the authors chose to carry on based on t tests instead?

3. The methodology indicates that multiple comparisons were carried out based the same samples. Was there an adjustment for type I error considered, to account for this? 

Author Response

Reviewer 2.

Well written manuscript and definitely well presented.

Response: We thank this reviewer for the complimentary comments!

I would ask for some clarifications regarding data analysis in the 3 following points.

  1. An examination of the supplementary material shows that the estimated p values were based on two-sided independent samples t tests assuming equal variances. Still, a descriptive approach shows that the variances in the two groups, especially in some indices, including the ones that statistically significant differences were detected, are rather different. Based on which criterion did the authors choose to carry out an analysis assuming equal variances in, unexceptionally, all differences examined?

Response: These is a good suggestion!  We have reanalyzed the data below as suggested using non-parametric T-tests (shown as new Table S1 in the revised manuscript). The criteria of significance are p < 0.05 and q < 0.25.

Pathways &   Metabolites

Mann-Whitney tests

Pathways & Metabolites

Mann-Whitney test

p value

q value

p value

q value

Ketone metabolites

Amino acid metabolites

3-hydroxybutyric acid

0.0022

0.0996

phenylalanine

0.0931

0.3568

4-hydroxybutyric acid

0.8182

0.9180

leucine

0.9372

0.9798

lysine

0.0931

0.3568

Cholesterol metabolite

tyrosine

0.1320

0.4338

cholesterol

0.0087

0.1991

tryptophan

0.4848

0.8578

deoxycholic acid

0.0281

0.3153

isoleucine

0.3095

0.7494

squalene

0.6667

0.8933

alanine

0.8182

0.9180

glycine

0.9372

0.9798

Hexoses

threonine

0.8182

0.9180

glucose

0.0411

0.3153

cysteine

0.4848

0.8578

fructose

0.0260

0.3153

serine

0.4848

0.8578

galactose

0.0649

0.3319

tagatose

0.0931

0.3568

Fatty acid metabolites

hexose

0.0649

0.3319

docosahexaenoic acid (DHA; 22:6)

0.6991

0.8933

levoglucosan

0.8182

0.9180

arachidonic acid (C20:4)

>0.999

>0.999

1,5-anhydroglucitol

0.6991

0.8933

linolenic acid (C18:3)

0.4848

0.8578

linoleic acid (C18:2)

0.6991

0.8933

Glucose metabolite

oleic acid (C18:1)

0.3095

0.7494

glucose

0.0411

0.3153

palmitoleic acid (C16:1)

>0.9999

>0.999

glucose-6-phosphate

0.1797

0.5165

palmitic acid (16:0)

0.8182

0.9180

fructose-6-phosphate

0.0411

0.3153

3-phosphoglycerate

0.6991

0.8933

Adenosine signaling

lactic acid

0.0649

0.3319

adenosine

0.1320

0.4338

inosine

0.4848

0.8578

Pentose phosphate pathway metabolites

adenosine-5-monophosphate

0.9372

0.9798

ribose-5-phosphate

0.3095

0.7494

ribulose-5-phosphate

0.3939

0.8578

Krebs cycle metabolites

citric acid

0.6991

0.8933

aconitic acid

0.6991

0.8933

isocitric acid

0.5887

0.8933

alpha-ketoglutarate

0.3939

0.8578

succinic acid

0.6991

0.8933

fumaric acid

0.1797

0.5165

malate

0.5887

0.8933

As shown below, the statistical significance did not show much difference when analyzed using parametric T-test or non-parametric Mann-Whitney test. Therefore, our conclusions remain unchanged.

Metabolites

T-test

Mann-Whitney

normality test (alpha=0.05)

Equal variance F-Test

q (B-H)

p value

p value

q (B-H)

3-hydroxybutyric acid (3HB)

0.0199

0.0005

0.0022

0.0931

Yes

0.3423

4-hydroxybutyric acid

0.8643

0.5325

0.8182

0.9021

Yes

0.2052

cholesterol

0.2748

0.0128

0.0087

0.1861

Yes

0.0785

deoxycholic acid

0.3206

0.0286

0.0281

0.2947

No

0.0034

squalene

0.9423

0.8450

0.6667

0.8842

No

0.0067

glucose

0.3206

0.0677

0.0411

0.2947

No

0.0046

fructose

0.3206

0.0387

0.0260

0.2947

No

0.0016

galactose

0.3206

0.0745

0.0649

0.3102

No

0.0038

tagatose

0.3206

0.0743

0.0931

0.3335

No

0.0117

hexose

0.5568

0.1424

0.0649

0.3102

No

0.0000

levoglucosan

0.8420

0.4895

0.8182

0.9021

Yes

0.1139

1,5-anhydroglucitol

0.9423

0.8244

0.6991

0.8842

Yes

0.1683

glucose-6-phosphate

0.8225

0.3065

0.1797

0.5150

No

0.0000

fructose-6-phosphate

0.8194

0.2375

0.0411

0.2947

No

0.0000

3-phosphoglycerate

0.8643

0.5427

0.6991

0.8842

Yes

0.2091

lactic acid

0.3206

0.0575

0.0649

0.3102

Yes

0.4433

ribose-5-phosphate

0.9295

0.6358

0.3095

0.7394

No

0.3750

ribulose-5-phosphate

0.8225

0.3043

0.3939

0.8470

No

0.0020

citric acid

0.8237

0.4309

0.6991

0.8842

No

0.0304

aconitic acid

0.9423

0.8598

0.6991

0.8842

No

0.2372

isocitric acid

0.8225

0.3826

0.5887

0.8842

No

0.0243

alpha-ketoglutarate

0.9423

0.9052

0.3939

0.8470

No

0.0454

succinic acid

0.9423

0.8418

0.6991

0.8842

Yes

0.0779

fumaric acid

0.8225

0.3698

0.1797

0.5150

Yes

0.3336

malate

0.8237

0.4597

0.5887

0.8842

Yes

0.2320

phenylalanine

0.3206

0.0403

0.0931

0.3335

Yes

0.0484

leucine

0.9612

0.9612

0.9372

0.9829

Yes

0.0898

lysine

0.3206

0.0564

0.0931

0.3335

Yes

0.1082

tyrosine

0.8194

0.2477

0.1320

0.4367

Yes

0.4601

tryptophan

0.8225

0.3810

0.4848

0.8687

Yes

0.2554

isoleucine

0.8237

0.4570

0.3095

0.7394

Yes

0.1984

alanine

0.9423

0.8490

0.8182

0.9021

Yes

0.3736

glycine

0.9423

0.9204

0.9372

0.9829

No

0.4268

threonine

0.9295

0.6230

0.8182

0.9021

Yes

0.0502

cysteine

0.9423

0.7116

0.4848

0.8687

No

0.3308

serine

0.8225

0.3584

0.4848

0.8687

No

0.1087

docosahexaenoic acid (DHA; 22:6)

0.9423

0.8989

0.6991

0.8842

Yes

0.0346

arachidonic acid (C20:4)

0.9423

0.8547

1.0000

1.0000

Yes

0.4955

linolenic acid (C18:3)

0.8237

0.4175

0.4848

0.8687

Yes

0.4154

linoleic acid (C18:2)

0.9295

0.6485

0.6991

0.8842

Yes

0.4090

oleic acid (C18:1)

0.8225

0.2967

0.3095

0.7394

Yes

0.2780

palmitoleic acid (C16:1)

0.9423

0.7586

1.0000

1.0000

Yes

0.2659

palmitic acid (16:0)

0.9423

0.9136

0.8182

0.9021

Yes

0.2278

adenosine

0.3117

0.04

0.1320

0.4338

No

0.0143

inosine

0.9322

0.62

0.4848

0.8578

yes

0.7737

adenosine-5-monophosphate

0.987

0.99

0.9372

0.9798

no

0.9229

  1. Since the sample size would not allow for reliable normal distribution testing, were non-parametric tests considered? And if so why did the authors chose to carry on based on t tests instead?

Response: We performed the nonparametric test as this reviewer suggested, and the statistical differences are similar to that measured using parametric T-test. Please see the Response 6.

  1. The methodology indicates that multiple comparisons were carried out based the same samples. Was there an adjustment for type I error considered, to account for this? 

Response: This is a good suggestion! We reanalyzed the data with Mann-Whitney multiple comparisons based on the metabolites present in the same group of animals, followed by Benjamini-Hochberg correction. Only changes of 3HB and cholesterol levels are statistically significant (p<0.05 and q<0.25). However, the other metabolite changes, such as glucose, fructose, galactose, tagatose and hexose occur in the same trend (Fig. 1E & Table S1). In particular, the rise in 3HB content, which is expected to increase the easier accessible substrate for the mitochondria, is consistent with the overall reduction of cellular hexose levels.